# The Advances and Limitations of the Determination and Applications of Water Structure in Molecular Engineering

**DOI:** 10.3390/ijms241411784

**Published:** 2023-07-22

**Authors:** Balázs Zoltán Zsidó, Bayartsetseg Bayarsaikhan, Rita Börzsei, Viktor Szél, Violetta Mohos, Csaba Hetényi

**Affiliations:** Department of Pharmacology and Pharmacotherapy, Medical School, University of Pécs, Szigeti út 12, 7624 Pécs, Hungary; zsido.balazs@pte.hu (B.Z.Z.); bayartsetseg704@yahoo.com (B.B.); rita.borzsei@aok.pte.hu (R.B.); szel.viktor@pte.hu (V.S.); mohos.violetta@aok.pte.hu (V.M.)

**Keywords:** drug design, docking, crystallography, electron microscopy, solvation, free energy

## Abstract

Water is a key actor of various processes of nature and, therefore, molecular engineering has to take the structural and energetic consequences of hydration into account. While the present review focuses on the target–ligand interactions in drug design, with a focus on biomolecules, these methods and applications can be easily adapted to other fields of the molecular engineering of molecular complexes, including solid hydrates. The review starts with the problems and solutions of the determination of water structures. The experimental approaches and theoretical calculations are summarized, including conceptual classifications. The implementations and applications of water models are featured for the calculation of the binding thermodynamics and computational ligand docking. It is concluded that theoretical approaches not only reproduce or complete experimental water structures, but also provide key information on the contribution of individual water molecules and are indispensable tools in molecular engineering.

## 1. Introduction

Water is a molecular jolly joker of a living nature. It is a main solvent in bulk solution and cellular interfaces and fills the void spaces in tissues (the mass of the human body is made up of ca. 60% water [1]). Water also acts as an active matrix component and is involved in the stabilization of the biomacromolecules mediating macromolecular interactions, e.g., in signaling pathways [2,3,4,5], and in the binding of small molecules to their target structures [6,7,8,9,10,11,12]. From a structural point of view, the role of water can be further classified. There are water molecules that form the bulk solvent accounting for 85% of the water content of a cell [13,14], and they might either be exchanged with bound waters or participate in the (de)stabilization of solute complexes. Buried water molecules stabilize solutes internally, giving 10% of the ‘dry mass’ of proteins. Water molecules also bridge between solute (macro)molecules and fill the void volumes of interaction interfaces [9,13,15]. They can form a hydration shell [16] that is either conserved or displaced upon ligand binding. If hydration shell water molecules are conserved upon ligand binding, they turn into bridges forming a static network of solute–water and water–water hydrogen bonds [13,15,17,18,19,20,21,22,23,24]. Such a static network is characterized by a low mobility and acts by stabilizing complexes. On the other hand, dynamic networks characterized by loosely bound water molecules with a high mobility participate in the complex destabilization or non-selective binding of various ligands.

In molecular engineering, the above structural roles of water can be translated into energetic contributions. For example, in a target–ligand interface (the main stage of drug design), conserved and leaving water molecules can be distinguished upon ligand binding [16,19,25,26,27,28,29]. Conserved waters tend to stay and form bridges in the target–ligand interface and are often referred to as ‘happy’ waters (Figure 1).

There are also ‘unhappy’ water molecules displaced by the drug molecule during its binding to the target. An unhappy water molecule might offer a possibility for the enthalpic optimization of ligand binding. An indirect, ‘unhappy’, water-mediated interaction between a ligand and a target might be enthalpically less favorable compared to the direct binding of a ligand to a target amino acid residue after the displacement of an ‘unhappy’ water molecule [30]. During the displacement of an ‘unhappy’ water molecule, it moves to the bulk, and this process has a favorable entropic contribution to the free energy change of the binding reaction (ΔG_b_) [28]. On the other hand, targeting ‘happy’ water molecules can be useful, as they often bridge between the target and ligand. A ligand can be optimized to participate in this bridging interaction by adding functional groups with a hydrogen bonding capacity to the drug molecule, providing a favorable enthalpic contribution to the ligand binding, because the additional hydrogen bonding capacity can form hydrogen bonds with ‘happy’ water molecules, as well as with hydrophilic target amino acids. In drug design, increasing the ligand interactions with happy water molecules and pushing unhappy water molecules away into the bulk (Figure 1) can therefore increase the ligand binding specificity [31,32,33] and affinity [33,34,35] to the target. Therefore, the importance of considering water molecules in the drug design process has been long recognized [10,36]. Besides ligand optimization, water molecules have been also utilized to improve docking results (See Section 5 for details [35,37,38]). 

Despite the above importance of water in drug design, the determination of the structure and energy contribution of the water networks in intra- and intermolecular interactions is challenging for both experimental and theoretical approaches [13,39,40]. Water molecules are often too mobile, as they can change their positions in the meantime of picoseconds [6] and get lost in the large electron density maps of proteins [39]. On the other hand, theoretical approaches often have a considerable computational cost, calculating all the possible interactions with water molecules in a large simulation box. The present review gives a brief account on the above limitations and advances of the recent experimental and theoretical methodologies for water structure and their applications in the context of molecular engineering focused on biomolecules and drug design.

## 2. Experimental Determination of Water Structure

The experimental methods of X-ray/neutron crystallography [41], cryo-electron microscopy (Cryo-EM) [42,43,44], and Nuclear Magnetic Resonance (NMR) spectroscopy [45,46] can be considered as the primary techniques for the determination of molecular structures at the atomic level. While these methods provide a solid background for establishing the structure–activity relationships [47] of biomolecules and their complexes, they face several challenges in the determination of hydration structures. Most of these difficulties come from the complexity (hydration layers interconnected with hydrogen-bonding networks) and high mobility (dynamic exchange of water molecules between the layers) of hydration structures. Plausibly, water molecules and other non-amino acid moieties, such as ligands, ions, or metals, are not included in the amino acid sequence that is otherwise key information for protein structure determination, indicating the order and covalent links between amino acids. 

More than half of the experimental structures published in the Protein Data Bank (PDB, [48,49]) contain at least one water molecule (Figure 2a). Crystallography is the most powerful technique for the exploration of the networks of several water molecules. Cryo-EM, NMR, and other methods can assign far fewer (often individual) water positions (Figure 2a).

However, crystallography provides only a static picture of the structure of solute molecules and their first surrounding water shell [50]. Moreover, the assignation of water positions in electron density maps gained via Fourier Transform from the crystal diffraction pattern is often complicated, even in the first shell. One of the methods is based on the low B factors, by which the waters bound to the protein surface or another water molecule located in the first hydration shell can be confidently identified [51]. In buried regions, such as binding pockets or active sites, even the third hydration layer can be resolved [50,52]. Problematic, partially ordered waters located mainly in the second hydration shell [53] can be assigned using D_2_O-H_2_O “neutron difference maps” [54]. This method uses the large difference in neutron scattering by deuterated and light waters, resulting in peaks of only water locations, while the scattering of the solute remains the same [51,53,54]. While neutron diffraction is capable of detecting not only oxygen but also hydrogen/deuterium atoms, and has been continuously developed [55], it is still less widespread due to the technical complexity of the method [56] and the limited accessibility of the neutron sources based on only four nuclear reactors worldwide [57], also reflected by the small number of structures [49] resolved by this method (Figure 2a). 

There are various computational methods that help with the assignation of water positions in electron density maps, sometimes equipped with quantum- and/or molecular mechanics refinements [58]. For example, PHENIX [59,60,61] is a frequently used system for macromolecular crystallographic structure solutions, in which a bulk-solvent determination protocol is based on both maximum-likelihood and least-squares target functions [62]. Coot performs a cluster analysis on a residual map to find water places [63]. The assigned waters are then checked based on their distance from the hydrogen-bond donors and acceptors, temperature factor, or electron-density level [63]. ARP/wARP [64] includes a fully automated placement for finding ordered water molecules using least-square refinement, in combination with the Fo–Fc difference of the electron density maps [65] (where Fo and F_C_ are the observed and calculated structure factor amplitudes), and geometric assumptions such as interatomic distances, angles, and van der Waals radii, etc. [66]. Despite the development of new assignation tools, the determination of correct water positions remains problematic, especially if the water structure is disordered surrounding non-polar atoms [67] or has fewer tetrahedral hydrogen bonds [68] at partially occupied solvent sites of low density. 

The number of water molecules determined by crystallography mainly depends on the size and form of the system [52], as well as its resolution. By increasing the size of the system, the number of water molecules [41] also increases and the solvent becomes considerably disordered. At the size of the proteins (MW > 30.000), the resolution is usually between 1.5 and 2.5 Å, having a high background noise level in Fourier maps due to the high incoherent scattering cross-section of the numerous hydrogen atoms [41], which makes the solute assignation more difficult [41,69]. Furthermore, in the case of large biomolecules, the number of hydration layers increases, resulting in a weaker and more diffuse solvent density [41]. The other problem is that the electron density of water molecules is similar to that of small iso-electronic ions (e.g., sodium and ammonium), leading to inaccurate assignation. Moreover, the experimental conditions also affect the successful, accurate, and valid water assignation. The limitations of crystallography include the difficulty of the crystallization of biomolecules, especially in the case of large, non-globular, or disordered systems [50]. Furthermore, it is also questionable how the crystallization procedure, such as packing and cryogenic temperature, modifies the native structure of the biomolecule [50] and its hydration shells. It has been proven that the hydration structure of a biomolecule highly depends on temperature [70]. 

In the case of cryo-EM, high-resolution structural information is gained from thousands of images produced by transmitting an electron beam through the protein sample embedded into a special vitreous environment instead of crystals. Thus, the biomolecules can be studied in a more “native” environment, with different conformational and/or functional states, and this allows for the resolution of structures in a higher molecular weight range than that of X-ray crystallography [71]. Due to the progressive improvement in technological and refining processes, the resolution of cryo-EM maps has been entered into the atomic dimension, where the resolvability of individual atoms, including solvent water atoms, is accessible [72]. The first cryo-EM structure with water molecules was published in 2003 (PDB code: 1uon) [73] at a resolution of 7.6 Å, which is too low for the identification of individual atoms. Due to the ‘Resolution Revolution’ [74,75], which started in 2013, less than a decade ago, when the first near-atomic resolution cryo-EM structures were published [76,77,78], the number of water-containing cryo-EM structures has exponentially increased (Figure 2b). This tendency might forecast that cryo-EM structures will catch up to the number of X-ray structures in the next decades, especially in the case of large protein complexes, cellular machines, and viruses [79]. The above computational methods used for the assignation of waters in X-ray crystallography could also be applied to cryo-EM maps. The development of new assignation tools has emerged in this field as well. The assignation of individual atomic positions in cryo-EM can be performed using methods such as SWIM (segmentation-guided water and ion modelling) [80] and UnDowser in MolProbity [81,82]. 

Unlike crystallography and cryo-EM, NMR spectroscopy is suitable for examining small proteins or oligopeptides in solutions adopting various conformations [50]. Water–protein interactions can be identified by using the nuclear Overhauser effect and/or rotating-frame Overhauser effect between the water protons and protein atom nuclei [83]. Here, individual water molecules can be determined that are located in the first hydration shell and bound to the protein instead of a complex 3D hydration structure [84]. It is notable that this method is limited by the short-term period of protein–water interactions, the hydrogen exchange with unstable protein moieties, and long-range dipole coupling, and identifies only 1–100 water molecules at best (Figure 2a). Additionally, while crystallography and cryo-EM provide direct information on the positions of water oxygen atoms, solution NMR is based on different principles.

## 3. Calculation of Water Structure

While there is an impressive, continuous development of experimental structure determination methods, the previous Section also highlighted the limitations of their assignment of the positions of water molecules [9,13,40]. To fill the gap of missing experimental hydration structures, extensive theoretical research has been conducted and resulted in various methods for the calculation of water positions (Table 1). 

Despite the flaws of these experimental methods, the validation of theoretical methods still relies on the experimental water oxygen positions. The positions of the predicted water oxygen and experimental water oxygen are compared, and if the distance is within a tolerance threshold, then it is accepted as a successfully predicted water oxygen position. The ratio of the count of the successfully predicted water oxygen positions and all the available experimental water oxygen positions can be considered as a success rate (SR, this number is expressed in percentage after multiplication with 100 in Table 1). The validation and comparison of different theoretical methods can be easily performed based on SR values (Table 1). 

Out of the four types (bulk, buried, interface, and surface) of water molecules mentioned in the Introduction (Figure 1), mostly surface and interface water molecules are investigated by theoretical methods (Table 1). Hydrated targets have surface water molecules in their first hydration shell [40] that have a stabilizing function on the macromolecular structure. Target–ligand complexes also have interface water molecules bridging between the target and ligand [13,15,19,25,106]. The prediction of interface water molecules can be accomplished very precisely with SR values even above 90% (Table 1) [9], as these molecules are captured between the target and the ligand, and there is enough space to fit only the water molecules participating in the interaction. Surface water molecules tend to have a higher mobility (B-factors) and can be predicted with SRs of ca. 80% (Table 1) [24,29,40]. That is, the natural uncertainty of surface water positions tends to result in a lower success of their prediction [39,40]. 

Either static or dynamic methods are used for the prediction of interface or surface water molecules. Static methods assume a static hydration shell and predict the binding sites of the water molecules on the surface of a dry experimental solute structure [40]. Finding a binding site can rely on energy calculations, scoring, prior knowledge, and information on H-bonds, and neural networks have also been applied [107]. Knowledge-based methods rely on the information found mainly in X-ray crystallographic structures (see previous Section). The main limitation of knowledge-based approaches is the assembly of an appropriate test set. The quality of X-ray crystallographic water oxygen positions varies greatly (see previous Section) and the methods perform better on similar structures that are involved in their test sets. Some methods assign a score to the experimental water molecules. Energy calculations may also apply popular docking tools to predict water molecule positions. Energy- or grid-based methods try to locate the energetically favorable positions of water molecules using probes that mimic them. Static methods can accurately identify the water molecules at the interfaces of the proteins and ligands, as these waters are usually static; however, a dynamic exchange of water molecules between the bulk solvent and the protein surface is disregarded by these methods. Generally, static methods do not consider an explicit water model and provide fast results. However, the quickness of these methods often involves a compromise in their precision. 

Dynamic methods rely on extensive molecular dynamics (MD) simulations or other global search techniques using an explicit water model and allowing for the mobility of individual water molecules. All atomic movements are recorded into trajectories and the protein–water, ligand–water, and water–water interactions can be followed. This includes a dynamic exchange of water molecules with the bulk solvent and the displacement of water molecules from the binding site by ligands. However, the analysis of each trajectory in a large-scale study using various systems would be time-consuming. To tackle this, the distribution density averages of the water molecules or their occupancy at binding sites might be investigated. Dynamic approaches offer information not only on the location of water molecules, but the displacement of water molecules can be also studied. MD-based thermodynamic analyses or a comparison of the hydration structures of the apo and holo targets can follow the application of these dynamic approaches. 

The counts of the systems and water molecules involved in the validation differ in different methods (Table 1). In future studies, the involvement of at least 1000 and 100 experimental (reference) water positions can be recommended for surface and interface predictions, respectively. Preferably, at least 10 different (protein or complex) systems should be used to have a diverse set of water positions. Notably, the SR depends on the choice of match tolerance, where the highest value is 2.5 Å, but more commonly 1.4–2.0 Å is used, which seems to be the consensus for method validation (Table 1). Naturally, when the match tolerance is set to a higher value (2.0–2.5 Å), the methods achieve better SRs. Notably, the calculated water positions and SR values correspond to a certain biomacromolecular structure (or PDB ID), and the use of high-resolution structures can be recommended for the calculation of the SR. While SRs provide a fair comparison of methods, the number of experimental water oxygen positions used for the method validation and testing is similarly important. Notably, MobyWat, WATGEN, and WarPP use more than 300 experimental water positions for the validation of interface hydration. Auto-SOL, AQUARIUS, Fold-X, and MobyWat use more than 1300 waters to test their surface predictions. 

The theoretical approaches of the above sections complement well the experimental methods for the atomic-level determination of water structures. In some cases, these methods also offer a complete hydration structure [13,40] of the protein surfaces and interfaces, which is often not achieved using experimental methods due to assignation problems (Section 2). Knowledge of the complete water structure is especially important for the calculation of (single molecular) the energy contribution of the (de)solvation process of drug–target binding (next Sections).

## 4. Water in the Structure-Based Calculation of Binding Thermodynamics

Water influences the thermodynamics of various biochemical interactions [108,109] important in molecular engineering. For example, ligand binding is described by binding free energy (ΔG_b_), a net measure of the strength of target–ligand interactions. During the formation of target–ligand complexes, hydration shells undergo considerable changes (Figure 1) and, therefore, the mediation of the interactions between the drug and target partners is fairly dependent on the water molecules. There are implicit and explicit water models for the calculation of the energetics of the (de)solvation during ligand binding. Implicit solvation models consider the solvent as a continuous medium around solutes and manifest in the formulae, e.g., in electrostatic terms [110]. Explicit water models place numerous water molecules in the simulation box and set various molecular properties for the water prototype used in copies [111,112]. Both types of models have been implemented at the molecular mechanics (MM) and quantum mechanics (QM) levels of calculations. 

At the molecular mechanics level, implicit water models such as MM-PB(GB)/SA [113] are widely used and based on the solution of the theoretically accurate, but computationally expensive Poisson–Boltzmann (PB) equation, or a simplified but scalable Generalized Born (GB) equation, to obtain the polar contribution of the solvation free energy change on an ensemble of MD snapshots. The solute cavity formation within the solvent and the van der Waals interactions between the solute and the solvent are represented by a nonpolar term often based on solvent-accessible surface areas (SA) [114]. Docking programs have also implemented implicit water models in their scoring functions due to their simple formulation and low computational costs. Notably, the scoring functions of docking methods require the fastest possible approaches to maintain their high-throughput nature. For example, the popular docking program AutoDock [115] applies the method of Stouten et al. [116], which calculates the solvation free energy as a sum of the atomic contributions with a linear relationship between the percentage of free volume around the atom and its contribution. At the same time, a PB-based distance-dependent dielectric function was also implemented in the Coulomb potential of AutoDock, which dampens the water permittivity value and corrects the screening effects near the solute surfaces [117]. In this way, a continuous transition of the relative permittivity of the medium is considered as we go from the bulk water to the protein surface. Similar implicit solvation terms have also been implemented in other popular docking software such as DOCK [118], MOE [119], and FITTED [120]. 

While implicit water models are useful for the approximation of long-range electrostatic forces considering the above-mentioned screening effect of solvent dielectric [110], they cannot handle hydration shells (Figure 1) and specific water-mediated linkages. However, the absence or presence of a certain water molecule at the binding site can drastically modify the overall affinity of ligands [121,122]. Thus, an accurate calculation of the binding thermodynamics is a rather impossible undertaking without the representation of individual water molecules. 

Explicit water models have been introduced to overcome the above-mentioned limitations of implicit approaches. At the MM level, there are various explicit models, such as SPC [123], TIP3P [124], and TIP4P [124], where the abbreviated names refer to the charge systems and sites (parameters) of the water molecule prototype. Explicit approaches allow for the calculation of the energy contributions of individual waters, e.g., using the statistical mechanical inhomogeneous fluid solvation (IFST, [125]) or grid inhomogeneous solvation (GIST, [126]) theory. In this way, the enthalpic and entropic terms of bound waters can be also considered, like in Wscore [127], DOCK-GIST [35], and AutoDock-GIST [38]. In some cases, the incorporation of explicit waters with the above methods has improved the correlation between the experimentally determined binding affinity and the docking score [38,127], while other works have not observed such improvements [87,128].

In the realm of quantum mechanics, theory permits a more accurate calculation of the charge distribution of molecules compared to MM. The assessment of the electrostatic interaction between the solute and water, in theory, can be included in the self-consistent field (SCF) calculation using dielectric continuum models [110]. However, for realistic solute cavities, it requires a numerical iterative process for every SCF cycle, which is extremely computationally demanding [129]. The Conductor-like Screening Model (COSMO) [130] solves this problem using a Green function description with analytical gradients, making the method practically applicable. COSMO can be considered as an advanced version of the polarizable continuum model (PCM, [131,132]), and is the most accurate implicit solvation model for semi-empirical QM. There is also a universal solvation model based on solute electron density (SMD, [133]), which is usually implemented for more computationally demanding levels of QM. At the semi-empirical level, the combination of PM6s [134,135,136,137] and PM7 [137,138,139] parametrizations, combined with the implicit model of COSMO, is a popular choice for estimating the binding affinities of ligands to targets.

Advances in computational speed and linear scaling methods [140] have allowed for the combination of implicit (COSMO) and explicit models handling long-range electrostatics and individual water contributions, respectively. Such hybrid approaches present a fast QM alternative of MM scoring functions for drug design. For example, Nikitina et al. inserted possible interface waters into hydrogen bond donor–acceptor sites and used the PM3 method [141,142]. Horváth et al. predicted interface waters using a molecular-dynamics-based method, MobyWat [40], and utilized the hybrid water model with PM7 parametrization for the estimation of the binding enthalpies (ΔH_b_) of ligands [143]. Here, the inclusion of explicit waters in the hybrid model yielded, e.g., a 3-fold smaller relative error when compared with vacuum calculations (Figure 3). Cavasotto et al. used a single-point PM7 calculation, keeping crystallographic waters in the binding interface in their QM docking scoring methodology to show encouraging enrichment factors on 10 protein targets [144]. The latter studies also extract the binding site environment from the target (similarly to Figure 3) to further reduce the computational time. The ΔG_b_ was also calculated by Hyslova et al., using a DFT-D3 and PM6-D3X4 combined method with crystallographic waters in the binding pocket, achieving a better fit with the combined implicit/explicit procedure (R^2^ = 0.68) compared to the implicit alone (R^2^ = 0.49) [145].

## 5. Water in Target-Ligand Docking

Target–ligand complex structures (Figure 1) are key to the engineering of new drugs. Computational docking can supply such atomic-resolution complex structures rapidly and, therefore, it is a widely used [146,147,148] alternative of experimental structure determination techniques (Section 2) in ligand screening projects [149,150,151]. Water molecules are active participants in real docking situations, as described in the Introduction (Figure 1). However, the proper use of these water molecules during computational docking is not trivial [36]. The inclusion of happy waters (Figure 1) bridging in the target–ligand interface may help to increase the precision of the docking calculation. On the other hand, if unhappy waters (Figure 1) are included in the interface, they would erroneously block the docking to the target sites used by the ligand in reality. Thus, the misuse of unhappy waters in an interface obviously leads to the mis-docking of the ligand. However, without knowledge of the true hydrated complex structure, it is rather difficult to distinguish between happy and unhappy water molecules in advance. Docking with waters is therefore a true “chicken and egg situation”, where let us say water is the chicken and the docked ligand is the egg. Docking is expected to produce a proper target–ligand complex for the decision on the inclusion of happy water molecules in (or the exclusion of unhappy ones from) the docking itself. This awkward situation is reflected in the corresponding literature. Several studies have reported that the incorporation of specific water molecules in the docking process significantly improved the docking performance [127,152,153,154], while others have found that including water molecules had little effect on this performance [155,156]. Several fast docking tools and strategies [38,92,97,127,157,158,159,160,161,162,163,164,165,166] have been developed to incorporate waters in the binding site during docking simulations. Many of these tools work with experimentally determined (known) water positions [160,163,164,165].

A simple way of incorporating water molecules into docking simulations is to include them as a static part of the target [167], where the positions and orientations of these water molecules are kept restrained during the simulation [162,167]. This strategy is used most in molecular docking studies and has been shown to be effective [168,169,170]. An improvement to the restrained water model is the displaceable water model, where the included water molecules can be switched on/off automatically during the simulation so that a ligand can keep the favorable water molecules and displace the non-favorable water molecules (GOLD, FlexX, FITTED, and DOCK). These included waters have mostly fixed positions or a limited mobility during the docking process, while some methods allow for the waters to change their positions and orientations through the search algorithm (in FITTED). 

Other methods solvate the ligand and then dock the solvated ligand with full flexibility, as the waters are kept or displaced depending on the entropy and/or energy contributions during the simulation (AutoDock4, MVD, and Glide). Such ligand-centric methods treat water molecules as a flexible part of the ligand, so they present the same flexibility as the ligand itself. RosettaLigand includes water movement both independently and dependently from the ligand during the initial stage, while considering the full flexibility of the target and the ligand through MC search. However, when the method was evaluated on a dataset of 341 diverse protein/ligand complexes from CSAR, no significant improvement was observed in the docking success rate [165]. This could be caused by there being no solvent-specific scoring adjustments in RosettaLigand other than the desolvation energy calculated using an implicit solvent model. Such a desolvation term in a force-field-based scoring function is often calibrated for protein–ligand complexes with no explicit solvent [165]. 

In many cases, the experimental water positions are not available or the hydration structure is not complete. In such situations, theoretical methods (Section 3) can supply the water positions for the docking calculations. Solvation before docking and a short molecular dynamics (MD) simulation were performed to improve these water positions, and after the encounter of the interacting partners, the water was removed based on a Monte-Carlo approach in HADDOCK [93,171] and a semi-explicit water model implemented in Rosetta [172]. This method improved the docking results for HADDOCK when compared to docking without explicit water molecules. FITTED [120] treats water molecules as a part of the target, and conserved water molecules are considered by an entropic penalty in the final score. This improves the docking accuracy for HIV-1 inhibitors. GOLD relies on two programs, FlexX [97,173] to pre-calculate the energetically favorable water sites and insert spherical water molecules (particles), and Consolv [174] to predict the water molecules that are likely to be displaced [166]. There is an upper limit of water molecules that is handled by this approach, so as not to increase the complexity and therefore the computational costs above a rational limit [36]. Although GOLD predicted the conservation or displacement of water molecules with a high efficiency, the effect on the ligand binding pose prediction was moderate [166]. Slide [160] enables the virtual screening of a relatively large set of ligands using Consolv [174] for the water prediction. 

HydroDock [175], a new approach, separates the chicken and egg situation and solves the hydrated docking using a parallel approach. The ligand is docked into the target without waters (dry docking). Simultaneously, the whole target is filled up with the explicit water molecules predicted by MobyWat [13,40]. Then, the dry docked complex and the hydrated target are merged and the water molecules that clash with the dry docked ligand are removed. The resultant complex is then energy minimized to set the proper orientation of the hydrogen bonds, and a short MD simulation is performed to yield the representative binding mode. HydroDock achieved a high accuracy in the case of ion-channel-bound ligand docking (Figure 4), one of the hardest cases of including water molecules in molecular docking [9,175]. As a specific example, the HydroDock method was validated on the ion channels of the influenza A and SARS-CoV-2 viruses.

As the inclusion of explicit water molecules increases computational costs [177], the scoring functions of many fast docking methods do not treat explicit water molecules with the proper partial charges and terms for their enthalpic or entropic contributions [177,178]. However, the way that water molecules are treated in the binding site and how their energetic contributions are evaluated is considered to be a key factor greatly affecting the docking performance [127]. To improve this docking performance, the contribution of water-mediated interactions and entropic effects may be considered for individual water molecules. A common modification to scoring functions is to add an entropy penalty, using a positive constant for each included water molecule to model the loss of rigid-body entropy favoring the displacement of the water molecules. However, in the case of large ligands, this approach leads to extremely positive energy contributions, necessitating a modification of the scoring function of AutoDock [179]. Moreover, Friesner and co-workers showed that the contributions of some water molecules to the free energy of binding can be much larger than others [180]. Some attempts have been made to calculate the target surface water sites and thermodynamics prior to the docking process, using third-party tools such as GIST and WaterMap, and to incorporate the solvation information in the scoring function (WScore, AutoDock-GIST, and DOCK-GIST). Although evaluation studies have reported only minor improvements in the success rate of docking for WScore and AutoDock-GIST, such an incorporation of explicit waters into the energy calculation definitely improved the correlation between the experimentally determined binding affinity and the docking score [38,127]. On the other hand, knowledge-based methods such as Consolv [174] (a k-nearest-neighbor-based classifier trained on 5542 molecules taken from 30 independently solved protein structures) can be used to determine the probability of the water molecules in the binding site to be conserved or displaced, as well as their corresponding desolvation penalty values (implemented in Slide) [160]. Instead of on-the-fly energy evaluations, scoring functions with more accurate desolvation functions can be implemented as re-scoring tools after the docking. For example, Wang et al. used molecular-mechanics–Poisson–Boltzmann-surface-area (MM–PBSA) re-scoring to find HIV-1 reverse transcriptase inhibitors [181], and several studies have reported that rescoring using a molecular-mechanics–generalized-Born-surface-area (MM–GBSA) method improved the enrichment of the known ligands for several enzymes and even the identification of substrates [182,183,184]. 

In the last decade, targeting protein–protein/DNA/RNA interactions has been considered to be a promising strategy for drug discovery [185,186,187,188], and a growing number of docking methods have been specifically developed for this [189]. Their shallow and relatively large interface (more than 1500 Å2 compared to the 300−1000 Å2 range for binding sites) [190] makes it readily accessible to the solvent or water-permeable in the case of nucleic acids. For nucleic acids, unlike proteins, their phosphate groups and corresponding counter ions (such as Mg^2+^ or K^+^) cause polarization upon the water molecules and functional groups of drugs. Thus, water molecules often play an important role in the ligand recognition and complex stabilization for nucleic acids, as well as proteins. There are several publications that have reported improvement in the success rate of the docking results when water molecules were included for RNA [191], DNA, and protein–protein complexes [192,193]. However, due to their large, solvent-accessible interface, it is extremely challenging to incorporate water molecules into the process of docking macromolecules within a reasonable computation time. Thus, the effect of these water molecules is often ignored in protein–protein/DNA/RNA docking, in which the desolvation penalty is estimated as proportional to the solvent-accessible surface area. There have been attempts to incorporate solvation effects in macromolecule docking. For example, HADDOCK explicitly treats water molecules by performing rigid-body docking on solvated macromolecules, followed by a Monte-Carlo (MC) simulation that displaces the waters based on their probabilities to form water-mediated contact, predicted using the Kyte-Doolittle scale [194]. Pavlovicz et al. developed a semi-explicit water model (implemented in Rosetta), in which a modified MC simulation displaces or adds explicit solvent molecules from bulk, followed by an energy evaluation with an implicit solvation energy term. Both attempts have improved the docking and ranking results [171,172].

## 6. Conclusions

Molecular engineering and drug design have been continuously fueled by the development of experimental structure determination techniques. However, the determination of the position of individual water molecules is often limited by the low resolution of their measurements. Theoretical calculations can supply the atomic-resolution hydration structure of target–ligand interfaces with a high precision, and often complement experimental techniques. The energetic contribution of individual water molecules to the full thermodynamics of target–ligand binding can be also calculated. There has been an improvement in the application of water structures in computational docking, a technique often used in the high throughput virtual screening of ligands in the drug industry. While “docking with waters” is still a problematic “chicken and egg situation”, a number of methods have been featured that answer this challenge as well.

## Figures and Tables

**Figure 1 ijms-24-11784-f001:**
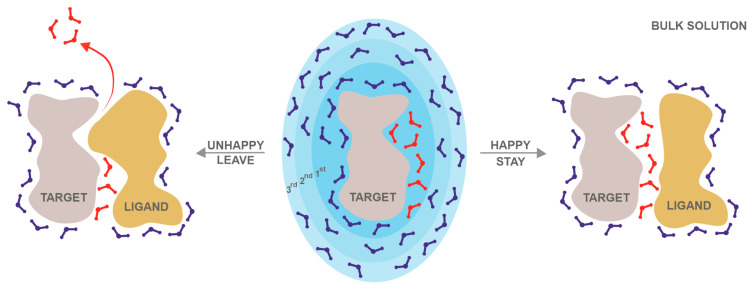
The hydration shells and the possible roles of interface water molecules during ligand binding. The target molecule (grey cloud, in the middle) is covered by hydration shells of surface water molecules (blue sticks), where the fading color of the shells represent the diminishing strength of interaction between the shell (also labelled by a serial number) and the target. Happy interface water molecules (red sticks, on the right) tend to stay, while unhappy water molecules (on the left) are displaced by the ligand (beige cloud) and leave (red arrow) to the bulk solution during the binding process.

**Figure 2 ijms-24-11784-f002:**
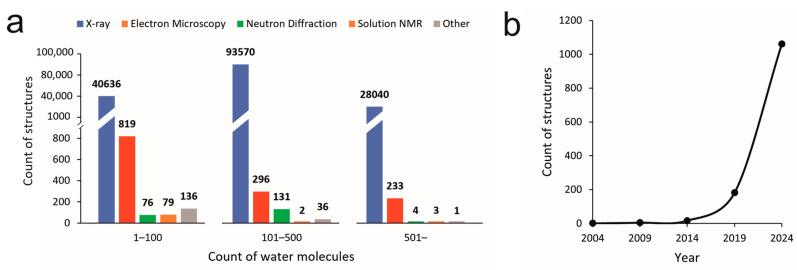
(**a**) Counts of structures containing water molecules resolved by different methods and deposited in the Protein Data Bank. Data were collected by the advanced search module of the PDB database: Entry features > Number of water molecules per deposited model. The number of structures was automatically separated by methods used for resolving. (**b**) The number of water-containing cryo-EM structures resolved in 5-year-long periods of time. The number on the x axis indicates the last year of the period.

**Figure 3 ijms-24-11784-f003:**
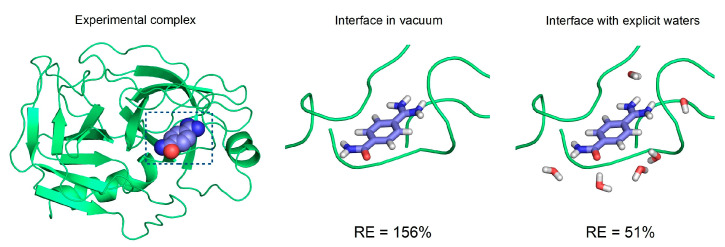
The complex structure of beta-trypsin (on the left, target in cartoon representation) and p-amidinobenzamidine (ligand marked with spheres). The target–ligand interface extracted for ΔH_b_ calculations is marked with a box. The close-up of the dry (middle) and explicitly hydrated (right, used for hybrid calculations) interface with a ligand in sticks representation. Relative errors (RE) of the calculated binding enthalpy (ΔH_b_) values of the dry and hybrid models are indicated below the corresponding interface structure. The RE values were calculated as RE(%) = 100 (ΔH_b,calculated_ − ΔH_b,experimental_)/ΔH_b,experimental_. The coordinates and ΔH_b_ values were re-used from a previous study [143] (Table 1 and Supporting Supplementary Table S5, β = 0). The incorporation of explicit water molecules in the ΔH_b_ calculation considerably reduced the RE in this case.

**Figure 4 ijms-24-11784-f004:**
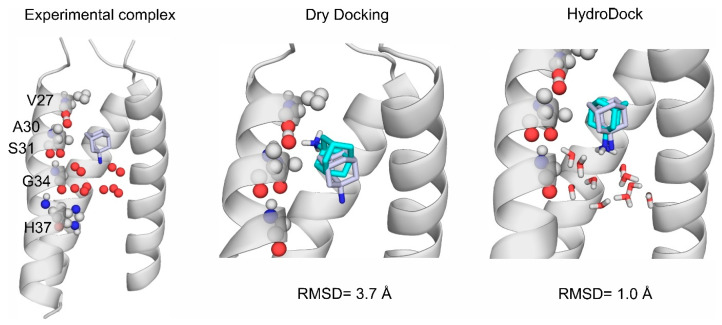
The role of water in ligand binding and the incorporation of explicit water molecules into docking using HydroDock. The influenza virus A ion channel is shown as grey cartoon, target amino acids are shown as spheres and labelled according to the 6bkk [176] PDB structure. Experimental water oxygen positions are shown as red spheres, and water molecules after HydroDock are shown as red and white sticks [175]. The experimental amantadine structure is shown as grey sticks, and the calculated structures as teal sticks. Dry docking (in the middle) fails to reproduce the experimental ligand binding mode, however, docking with water molecules (on the right) improved similarity with experimental results greatly. Root mean squared deviation (RMSD) is calculated after superimposition of the calculated to the experimental structure, between ligand heavy atoms.

**Table 1 ijms-24-11784-t001:** The categorization and performance of theoretical methods of prediction of hydration structure.

Method	Concept	Type ^a^	#System/#Water ^b^	Match Tolerance (Å)	SR (%)
3D-RISM ^c^ [85,86,87]	Knowledge	IF	18/113 ^d^	2.5	91
		IF	13/113 ^e^	1.5	65
		SF	8/101 ^e^	1.5	60
AcquaAlta ^c^ [88]	Geometry	IF	20/77	1.4	76
Auto-SOL ^c^ [89]	Geometry	SF	5/1337	1.5	64
AQUARIUS ^f^ [90]	Knowledge	SF	7/1376	1.4	59
Fold-X ^c^ [91]	Energy	SF	74/2687	1.0	76
Forli et al., 2012 ^c^ [92]	Geometry ^g^	IF	27/51	2.0	96
HADDOCK ^c^ [93]	Geometry ^g^	IF	27/50	2.0	90
Huggins and Tidor, 2011 [94]	Geometry	IF	5/19	2.0	68
HydraMap ^c^ [95]	Dynamic	IF	13/113 ^e^	1.5	72
		SF	8/101 ^e^	1.5	69
HyPred ^f^ [96]	Dynamic	SF	3/233	1.0	12
MobyWat ^c^ [13,40]	Dynamic	SF	20/1500	1.5	80
		IF	31/344	1.5	90
Particle concept ^h^ [97]	Geometry	IF	200/232	1.5	35
Splash’Em ^c^ [98]	Knowledge	IF	91/230	1.0	62
SZMAP ^h^ [99]	Knowledge	IF	18/113 ^d^	2.5	96
WaterDock ^c^ [100]	Energy	SF	7/92	2.0	88
WaterFLAP ^h^ [87,101,102]	Knowledge	IF	18/113 ^d^	2.5	98
WaterMap ^h^ [37,87]	Dynamic	SF	1/11	1.5	82
		IF	18/113 ^d^	2.5	96
WarPP ^c^ [103]	Geometry	IF	1500/20,000	1.0	80
WATGEN ^f^ [104]	Geometry	IF	126/1264	2.0	88
WATsite ^c^ [95,105]	Dynamic	IF	13/113 ^e^	1.5	75
		SF	8/101 ^e^	1.5	77

^a^ Water molecules in the target–ligand interface (IF), and on unbound target surface (SF) are considered, respectively. ^b^ The count of systems/the count of experimental water oxygen positions used in the cited study for method validation. ^c^ Freeware or free trial for academic use. ^d^ These data are taken from the comparative paper [87]. ^e^ These data are taken from the paper [95]. ^f^ Website no longer available. ^g^ With docking search. ^h^ Commercially available.

## Data Availability

Data sharing not applicable No new data were created or analyzed in this study. Data sharing is not applicable to this article.

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
