# Peer review of "The Advances and Limitations of the Determination and Applications of Water Structure in Molecular Engineering"

_ijms, 2023, doi:10.3390/ijms241411784_

Round 1

Reviewer 1 Report

This paper by the group of Csaba presented a comprehensive review on the determination and applications of water structure in molecular engineering, mainly focusing on the advances and limitations for receptor-ligand interactions in drug design. Experimental and computational methods were demonstrated for the determination of water structure, followed by water in the calculations of thermodynamic energetics between the binding partners and in the receptor-ligand docking. The review was well rewritten and organized in a logical way. This reviewer recommended its publication in the International Journal of Molecular Sciences after addressing the minor issues below.

(1) Lines 177-184: NMR spectroscopy was only suitable to determine the structure of small proteins or oligopeptides. Please comment on this.

(2) In section 5, the authors noted the necessity of a short MD simulation. Is it due to the computational cost of long simulations or the issues in the accuracy of MM force fields?

(3) A bridging water between metal ions and substrate may function as nucleophilic attack. In such case, determination of water position needs to consider the interaction with the bound metal ion. Any comments on this was very welcome.

Author Response

Reviewer 1

This paper by the group of Csaba presented a comprehensive review on the determination and applications of water structure in molecular engineering, mainly focusing on the advances and limitations for receptor-ligand interactions in drug design. Experimental and computational methods were demonstrated for the determination of water structure, followed by water in the calculations of thermodynamic energetics between the binding partners and in the receptor-ligand docking. The review was well rewritten and organized in a logical way. This reviewer recommended its publication in the International Journal of Molecular Sciences after addressing the minor issues below.

(1) Lines 177-184: NMR spectroscopy was only suitable to determine the structure of small proteins or oligopeptides. Please comment on this.

The authors agree with the statement of Reviewer 1, and inserted a comment on suitability of NMR for small proteins or oligopeptides.

(2) In section 5, the authors noted the necessity of a short MD simulation. Is it due to the computational cost of long simulations or the issues in the accuracy of MM force fields?

The short MD simulation was introduced to HydroDock protocol in order to allow the movement of the dry docked ligand, which is inaccurately placed in the absence of water molecules. The shortness (ca. 40 ns) of the simulation can reduce computational cost and it was also found sufficient for the goal of the study.

(3) A bridging water between metal ions and substrate may function as nucleophilic attack. In such case, determination of water position needs to consider the interaction with the bound metal ion. Any comments on this was very welcome.

The authors agree with the comment of the Reviewer. For example, the MobyWat method (also developed by our group) can consider water molecule interaction with bound metal ions while making the surface or interface water position predictions. The method is freely available at http://www.mobywat.com/index.php/download.

We acknowledge the Reviewer for the constructive evaluation of the manuscript.

Reviewer 2 Report

The submitted review is quite concise, although the number of references is large. The scientific level is very high, it is obvious that the authors are expert in this area. I appreciate this review as it would help researchers to answer the common question that they can be asked while presenting the results of molecular docking during a conference – “What about water?”. However, there are also some points that need to be clarified.

At some point, in abstract and introduction, it should be stated that this review is focused on the role of water in the structure of (bio)macromolecules. The Authors don’t mention the role of water in solid hydrates, which are very important from the molecular engineering point of view. Those waters, present in relatively small organic and inorganic crystals are much better characterized by the experimental methods mentioned in this manuscript, mostly due to their limited mobility.

Line 30-31, I can’t agree with this. After all water-biomolecule is a kind of intermolecular interaction.

Line 44, “happy” is quite unusual adjective to describe a chemical molecule, namely water. It is a bit too colloquial as well.

Line 117, it should be “D2O” and “H2O”

Lines 177-185, the other aspect, not mentioned here, is that while Cryo-SEM or crystallographic methods directly “detect” the positions of water Oxygen atoms, the solution NMR is quite different in its principles

Lines 202-210, I wonder what about the cases in which multiple structures of the same protein are present in the PDB, and while the conformations are more or less similar, the structures differ in the number and positions of water molecules. This will affect the SR, won’t it? Also, I think that this aspect (comparing the structure of the same biomacromolecule in terms of the presence of water) should be discussed in a more detailed way in this review.

Table 1, it would be beneficial if the authors include the information whether the chosen program is a freeware (at least for academic) or commercial

Lines 324—350, what about the other popular methods like PCM or SMD?

Also, the other aspect that should be mentioned is the influence on the concentration of molecule (and thus the macromolecule:water molar ratio) on the presence and structure of water.

I also wonder how the presence of common ions, such as Na+, Cl-, etc. affects the results of calculations of water positions.

Author Response

Reviewer 2

The submitted review is quite concise, although the number of references is large. The scientific level is very high, it is obvious that the authors are expert in this area. I appreciate this review as it would help researchers to answer the common question that they can be asked while presenting the results of molecular docking during a conference – “What about water?”. However, there are also some points that need to be clarified.

At some point, in abstract and introduction, it should be stated that this review is focused on the role of water in the structure of (bio)macromolecules. The Authors don’t mention the role of water in solid hydrates, which are very important from the molecular engineering point of view. Those waters, present in relatively small organic and inorganic crystals are much better characterized by the experimental methods mentioned in this manuscript, mostly due to their limited mobility.

The following (change highlighted with green background) is inserted into the abstract (lines 9-12): „While the present review focuses on target-ligand interactions in drug design with a focus on biomolecules, the methods and applications can be easily adopted to other fields of molecular engineering of molecular complexes including solid hydrates.”.

And in lines 80-83 of Introduction: „The present review gives a brief account on the above limitations and advances of recent experimental and theoretical methodologies on water structure and their applications in the context of molecular engineering focusing on biomolecules and drug design.

Line 30-31, I can’t agree with this. After all water-biomolecule is a kind of intermolecular interaction.

We agree. The sentence was deleted.

Line 44, “happy” is quite unusual adjective to describe a chemical molecule, namely water. It is a bit too colloquial as well.

The authors agree that this nomenclature might seem surprising at first, however the terminology was applied as early as 2012 by Mason et al (Trends in Pharmacological Sciences, 2012, 33, 5, 249-260) and also by others later.

Line 117, it should be “D2O” and “H2O”

Changed.

Lines 177-185, the other aspect, not mentioned here, is that while Cryo-SEM or crystallographic methods directly “detect” the positions of water Oxygen atoms, the solution NMR is quite different in its principles

The following is added to lines 185-7: „Additionally, while crystallography and cryo-EM give direct information on positions of water oxygen atoms, solution NMR is based on different principles.

Lines 202-210, I wonder what about the cases in which multiple structures of the same protein are present in the PDB, and while the conformations are more or less similar, the structures differ in the number and positions of water molecules. This will affect the SR, won’t it? Also, I think that this aspect (comparing the structure of the same biomacromolecule in terms of the presence of water) should be discussed in a more detailed way in this review.

If multiple structures of the same protein are available in the PDB, the structure with the best resolution should be selected for SR calculation, as resolution is an experimental evaluation metric of the quality of the structure (especially in terms of water oxygen placement). Accordingly, the following sentence was inserted at lines 265-8: “Notably, the calculated water positions and SR values correspond to a certain biomacromolecular structure (or PDB ID), and the use of high resolution structures can be recommended for calculation of SR.”

Table 1, it would be beneficial if the authors include the information whether the chosen program is a freeware (at least for academic) or commercial

Footnotes c (Freeware or free trial for academic use), f (Website no longer available) and g (Commercially available) were introduced to Table 1.

Lines 324—350, what about the other popular methods like PCM or SMD?

A few lines on PCM and SMD were inserted: “COSMO can be considered as an advanced version of the polarizable continuum model (PCM, [131,132]), and is a most accurate implicit solvation model for semi-empirical QM. There is also a universal solvation model based on solute electron density (SMD, [133]) usually implemented for computationally more demanding levels of QM.

Also, the other aspect that should be mentioned is the influence on the concentration of molecule (and thus the macromolecule:water molar ratio) on the presence and structure of water.

For drug design purposes the interaction of one biomacromolecule with a single ligand is interesting, and the present manuscript is focused on the determination of the role of water molecules in this particular aspect.

I also wonder how the presence of common ions, such as Na+, Cl-, etc. affects the results of calculations of water positions.

The effect of ions are partly included in MD-based methods like MobyWat, as for MD simulations (PME approach) the net charge of the system is zeroed by some counter-ions, and the predicted water oxygen positions are based on clustering the frames of a MD simulation trajectory.

We acknowledge the Reviewer for the constructive evaluation of the manuscript.

Reviewer 3 Report

The manuscript by Hetenyi and co-workers is an interesting survey of experimental methods and computational approaches that allow the determination of hydration waters surrounding biological macromolecules, in particular globular proteins. The main aim is to describe the procedures adopted to include water molecules in the binding region of protein-ligand complexes in order to arrive at accurate and reliable docking predictions. The manuscript is well organized and clearly written. The English is good, even though the use of the word “assignation” is not strictly correct.

I favor publication of the manuscript. At the same time, I think that the authors could provide more details on a specific example to improve the general understanding of the matter.

The English is in general good, but some editing can be useful.

Author Response

Reviewer 3

The manuscript by Hetenyi and co-workers is an interesting survey of experimental methods and computational approaches that allow the determination of hydration waters surrounding biological macromolecules, in particular globular proteins. The main aim is to describe the procedures adopted to include water molecules in the binding region of protein-ligand complexes in order to arrive at accurate and reliable docking predictions. The manuscript is well organized and clearly written. The English is good, even though the use of the word “assignation” is not strictly correct.

I favor publication of the manuscript. At the same time, I think that the authors could provide more details on a specific example to improve the general understanding of the matter.

The following is added to lines 446-447: „ As a specific example, HydroDock method was validated on the ion channels of influenza A and SARS-CoV-2 viruses.” Figure 4 shows the specific case of applicability of HydroDock on the ion channel of influenza A virus to reproduce the experimental binding modes of ion channel inhibitors, two of which are approved durgs for the treatment of influenza infection.

We acknowledge the Reviewer for the constructive evaluation of the manuscript.

Round 2

Reviewer 2 Report

The Authors have included my comments in the revised version of this work. This manuscript can be published as it is now.